# Testosterone and Cortisol Responses to HIIT and Continuous Aerobic Exercise in Active Young Men

**Cristian Cofré-Bolados [1,2,\*], Patricia Reuquen-López [3], Tomas Herrera-Valenzuela [1,4]**,
**Pedro Orihuela-Diaz [3,5], Antonio Garcia-Hermoso [1] and Anthony C. Hackney [6]**

[1]  Laboratory of Sciences of Physical Activity, Sport and Health, Faculty of Medical Sciences, Universidad de Santiago de Chile, Santiago 9170022, Chile; tomas.herrera@usach.cl (T.H.-V.); antonio.garcia.h@usach.cl (A.G.-H.)

[2]  Department of Physical Education, CEA-YMCA Adapted Exercise Center of Santiago, Santiago 9170022, Chile

[3]  Laboratorio de Inmunología de la Reproducción, Facultad de Química y Biología, Universidad de Santiago de Chile, Santiago 9170022, Chile; patricia.reuquen@usach.cl (P.R.-L.); pedro.orihuela@usach.cl (P.O.-D.)

[4]  School of Sports and Physical Activity Sciences, Faculty of Health, Universidad Santo Tomás, Santiago 9170022, Chile

[5]  Centro para el Desarrollo de la Nanociencia y la Nanotecnología CEDENNA, Santiago 9170022, Chile

[6]  Department of Exercise & Sport Science, University of North Carolina Chapel Hill, Chapel Hill, NC 27599, USA; ach@email.unc.edu

[\*]  Correspondence: cristian.cofre@usach.cl

**Abstract:** It is well known that physical exercise modifies plasma levels of testosterone and cortisol. However, the effect of high-intensity interval training (HIIT) on the plasma hormone levels is controversial. The aim of the study was to compare the effects of HIIT exercise or submaximal continuous aerobic exercise on circulating testosterone and cortisol levels in active male collegiate students. Methods: Thirteen moderately-active young adult males (20.2 (SD 2.1) years old) completed a HIIT (20 min of 15 s intervals of running at 110% of maximum oxygen consumption interspersed with 15 s of active rest at 40% of maximum oxygen consumption) and a continuous aerobic exercise (AEE) 20 min at 70–75% of maximum oxygen consumption. The mean total workload of both protocols was the same for each exercise session. Blood samples were collected pre-session (rest), immediately after the session (0 h), and 12 h post-session (12 h). Results: Both exercise protocols, similarly increased plasma levels of free testosterone immediately post-exertion ($p < 0.05$ AEE and $p < 0.01$ HIIT). No differences were observed between the conditions in the concentration of testosterone at 12 h. Cortisol level and Testosterone/Cortisol (T/C) ratio remained constant for all measurements, regardless of the type of exercise performed. Conclusion: The testosterone concentrations rose significantly post intervention in both HIIT and AEE condition, but 12 h post intervention there was no difference between conditions, decreasing to baseline (pre-intervention). The T/C ratio decreased significantly (below baseline) only in the HIIT condition 12 h post intervention.

**Keywords:** anabolic hormone; catabolic hormone; high intensity interval; steady-state aerobic exercise

## 1. Introduction

Physical exercise is a powerful stimulus for the endocrine system, modifying the plasma concentration of many hormones, including the steroid testosterone and cortisol [1]. Testosterone is a major anabolic hormone, while cortisol produces generally catabolic effects [2]. Thus, the testosterone/cortisol ratio (T/C) is considered a good indicator of anabolic/catabolic status in individuals [3]. Evidence shows that plasma variation of testosterone and cortisol occurs in response to

continuous aerobic exercise (AEE); i.e., during AEE, plasma concentrations of both hormones increase without modification of the T/C ratio [4] and this increase is proportional to the intensity of physical exercise [5]. Among other factors, the response of these hormones to exercise depends on the degree of training and hydration status of individuals [6,7].

Testosterone and cortisol levels after physical exercise also depend on the duration and intensity of exercise [6]. For example, Trembly et al. [8] reported changes to these hormones at rest and after 40 min, 80 min, or 120 min of AEE (50–55% of maximal oxygen uptake (VO2 max)), specifically free testosterone levels decreased by 10% and cortisol levels increased by 22% compared to values obtained before exercise and after 120 min. On the other hand, 10 min of high intensity exercise increases free testosterone levels after exercise without modifying free cortisol levels, and therefore increasing the T/C ratio [9]. This work suggests that both the volume as well as the intensity of physical activity determine the body's anabolic environment. However, because different studies use different total loads of physical activity, it is difficult to evaluate the actual impact of exercise duration and intensity on plasma levels of testosterone and cortisol. There is still no clarity regarding the effects of high-intensity training on plasma levels of free testosterone and cortisol during the recovery period after exercise in trained and untrained individuals.

High-intensity interval training (HIIT) is a training method used in trained as well as in untrained individuals. Intensity within this training usually corresponds to the third phase of the three-phase model above the anaerobic threshold, generally about 80–85% VO2 max [10], producing central adaptations associated with sympathetic activation and cardiovagal-adrenal improvement [11]. HIIT produces similar or higher cardiometabolic gains in the short term compared to continuous aerobic exercise of similar intensity [12], presenting a lower effort perception, lower levels of plasma catecholamines and less elevation of blood lactate concentrations [13]. The use of HIIT training with short durations, in efforts and pauses usually less than 1 min or at distances of 30 to 40 m [11] is common in runners and swimmers. HIIT training has recently gained much attention because of its effectiveness in triggering rapid adaptations in skeletal muscle metabolism and increases in maximum VO2 [14]. The effects of HIIT exercise on plasma testosterone and cortisol levels after exercise, however, are still unclear [12].

In this context, the effect of one bout of HIIT or AEE on free testosterone and cortisol levels was examined in active male collegiate students.

## 2. Materials and Methods

### 2.1. Subjects

Male undergraduate subjects (*n* = 13) studying physical education with a VO2 max relative of 50.9 mL/kg/min (7.8 SD) and whom self-reported more than 150 min per week of moderate-to-vigorous physical activity participated in the current study. All subjects gave their informed consent. The study protocol was approved by the University of Santiago Ethics Committee and complied with the principles of the Declaration of Helsinki.

### 2.2. Intervention for Exercise Sessions

Each subject performed a maximum incremental exercise test with CORTEX Metamax® 3B Cortex Biophysik GmbH Leipzig, Germany metabolic equipment. The protocol used was the American College of Sports Medicine (ACSM) incremental speed test [15]. It started at 6.4 km/h, and speed increases 1.6 km/h (1 mile/h) every minute. This procedure aimed to determine VO2 max and speed of maximum oxygen consumption (vVO2 max). The maximum HR and an RER 1.1 were used as maximality criteria. Three days after the test, the subjects showed up at 19:30 h in the laboratory and during three non-consecutive sessions, separated by 72 h, they performed the control sessions (CON), HIIT, and AEE. The training sessions consisted of a warm-up of 5 min walking at 5 km/hour and 20 min on a motorized treadmill. During AEE sessions, the treadmill speed remained between 70% and 75%

vVO2 max. HIIT was performed in 15 s intervals of running at 110% vVO2 max on an interval and 15 s at 40% vVO2 max on a second interval of active recovery (treadmills placed next to each other) during a 20 min period. We sought to balance the total physical exercise load between AEE and HIIT. Three blood samples were collected, the first one before exercise (rest), the second immediately after the exercise protocol AEE or HIIT (0 h) and the third 12 h after exercise AEE or HIIT (12 H). For the control session, the protocol consisted of staying in the supine position for 20 min. The next morning the subjects arrived at the laboratory at 8:00 a.m. for a fasting 12 h. sample. The second and third blood samples were taken after 7–8 h of sleep. After the blood draw a standardized snack (a cereal bar and a box of 200 mL juice) was provided. Prior to each session and measurement, subjects were asked to refrain from exercising beyond what was required for the study, were asked to be free from emotional stress, and sexual activity was prohibited for 24 h. A standardized diet (approximately 60% carbohydrates) was recommended during the sampling and protocol application process.

### 2.3. Procedure for Blood Sample

Blood was collected using venipuncture procedures (antecubital vein) using a 10cc syringe equipped with a 2.5 cm needle. All blood was immediately transferred to an EDTA treated Vacutainer® tube and placed on ice until further processing. Each whole blood sample was centrifuged at $2000\times g$, 4 °C and stored at −20 °C until biochemical analysis. Hormone measurements consisted of free levels of testosterone and cortisol in blood plasma. This measurement was determined by immunoreactivity-activated procedures (chemiluminescence) at Sanasalud Laboratories, Santiago, Chile (www.sanasalud.cl).

### 2.4. Statistics

Results were expressed as mean ± standard deviation. To compare free testosterone and total cortisol levels in blood between AEE and HIIT conditions at different times, a repeated-measures analysis of variance (ANOVA) with Dunnet posthoc tests were used. To ensure the validity of ANOVA analyzes, the Levene test for homogeneity of variance and Greenhouse–Geisser of sphericity adjustments were applied. Additionally, the effect size (r2) was calculated as the SS (sum-of-squares) treatment divided by the sum of the SS treatment plus the SS residual. A significance level of $p < 0.05$ was established. Statistical analyses were performed using Statistical Package for the Social Sciences software for Windows version 21.0 (IBM Corporation, New York, NY, USA).

## 3. Results

Descriptive statistics are shown in Table 1.

**Table 1.** Characteristics of the sample.

| Characteristics | Mean | SD |
|---|---|---|
| Age (years) | 20.2 | 2.1 |
| Height (cm) | 170.6 | 4.0 |
| Weight (Kg) | 72.6 | 2.5 |
| Percentage of fat | 15.9 | 2.3 |
| VO$_2$ max (L/min) | 3.7 | 0.29 |
| VO$_2$ max (ml/kg/min) | 50.9 | 7.8 |

As shown in Figure 1, no differences were observed between conditions in plasma cortisol concentrations (Control = 8.16 ± 3.3; AEE = 10.13 ± 4.5; HIIT = 8.75 ± 3.3 nmol/L; $p = 0.262$; r2 = 1.06), testosterone (Control = 13.47 ± 6.3, AEE = 13.47 ± 2.9, HIIT = 13.55 ± 3.8 nmol/L; $p = 0.994$; r2 = 0.000) or T/C ratio (Control = 1.94 ± 1.2, AEE = 1.66 ± 1.0, HIIT = 1.82 ± 0.9; $p = 0.600$; r2 = 0.041) prior to physical exercise.

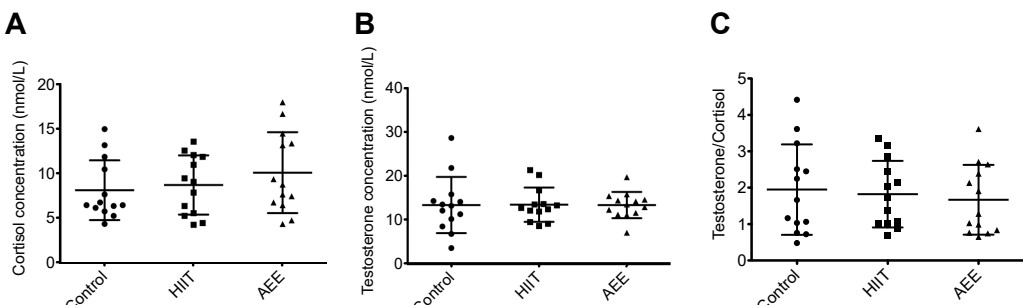

**Figure 1.** Levels of free cortisol, free testosterone and testosterone/cortisol ratio at rest. The levels of testosterone (T), cortisol (C), and the T/C ratio in the sample were determined prior to the realization of the different protocols (control, SIT and AEE). (**A**) plasma cortisol levels; (**B**) plasma testosterone levels; (**C**) Testosterone/Cortisol ratio. Data shown as mean ± SD.

Cortisol levels remained constant between rest and time 0 in the control condition (Figure 2A). However, it increased significantly at 12 h (8.16 ± 3.3 and 13.17 ± 3.4 nmol/L for rest and 12 h respectively; $p = 0.000$; r2 = 0.666) (Figure 2A). Plasma levels of cortisol did not change after AEE, reaching values of 10.13 ± 4.5; 11.95 ± 5.6 and 13.99 ± 2.1 nmol/L for resting conditions, 0 h and 12 h after exercise, respectively, although a strong trend was observed for the increase of free cortisol levels at 12 h post AEE ($p = 0.030$; r2 = 0.265) (Figure 2B). The HIIT exercise protocol increased plasma cortisol levels 12 h post-intervention compared with rest values (8.75 ± 3.3 and 13.25 ± 2.2 nmol/L for rest and 12 h respectively; $p = 0.000$; r2 = 0.414) (Figure 2C). No differences were observed in any of the measurements of free cortisol levels between the AEE and the HIIT condition.

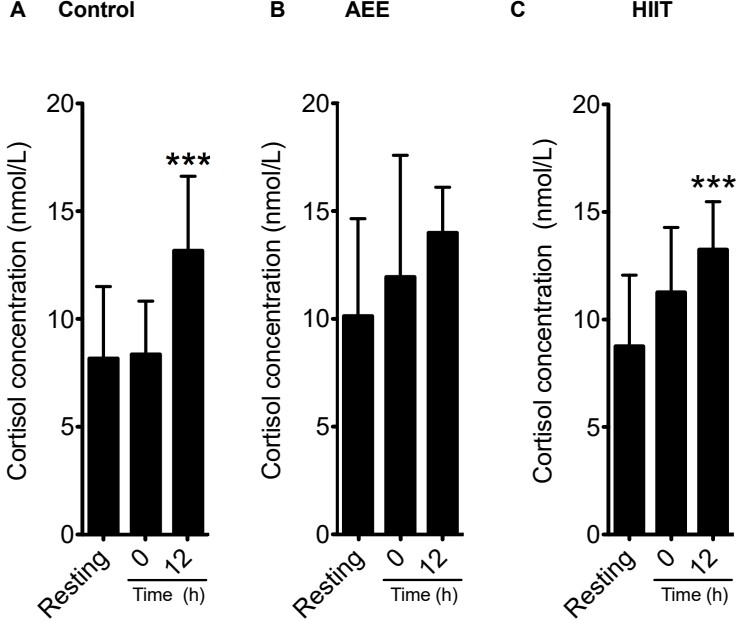

**Figure 2.** Effect of different exercise protocols (AEE and HIIT) on plasma levels of free cortisol. Resting cortisol levels, immediately post-intervention (0 h) and 12 h post-intervention (12 h) were determined. (**A**) Control condition; (**B**) AEE condition; (**C**) HIIT condition. *** $p < 0.001$ versus rest. Data shown as mean ± SD.

Figure 3 shows the effect of different exercise protocols (AEE and HIIT) on plasma levels of free testosterone. Free testosterone remained constant in all measurements in the control condition ($p = 0.258$; r2 = 0.106) (Figure 3A). However, both AEE and HIIT exercise protocols showed an increase in plasma testosterone concentration to a peak immediately after exercise, from 13.47 ± 2.9 nmol/L at rest

to 16.53 ± 3.1 nmol/L for the AEE condition and from 13.55 ± 3.8 nmol/L at rest to 17.92 ± 4.4 nmol/L for the HIIT condition. Returns to resting values 12 h post intervention was observed for both conditions ($p = 0.000$; r2 = 0.574, Figure 3B and $p = 0.000$; r2 = 0.620, Figure 3C). There were no significant differences in the magnitude of these changes when comparing conditions.

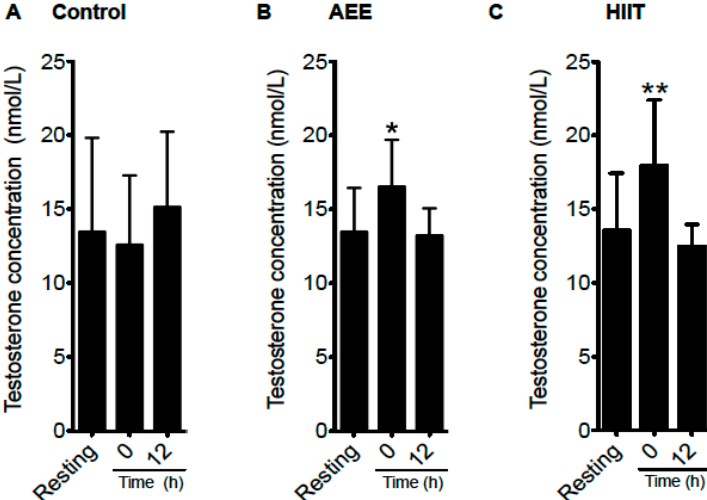

**Figure 3.** Effect of different exercise protocols (AEE and HIIT) on plasma levels of free testosterone. Resting testosterone levels, immediately post-intervention (0 h) and 12 h post-intervention (12 h), were determined. (**A**) Control condition; (**B**) AEE condition; (**C**) HIIT condition. * $p < 0.05$; ** $p < 0.01$ versus rest. Data shown as mean ± SD.

Finally, T/C ratio was used as an index of the anabolic environment of the participants before and after an AEE or HIIT exercise session. The T/C ratio was similar in all measurements performed in the control condition ($p = 0.232$; r2 = 0.331) (Figure 4A). Similar results were observed in AEE ($p = 0.142$; r2 = 0.322) (Figure 4B). In the HIIT, T/C ratio decreased after physical activity from 1.82 ± 0.90 at rest to 0.97 ± 0.23 at 12 h after exercise, with no difference between rest and 0 h post HIIT ($p = 0.002$; r2 = 0.401) (Figure 4C).

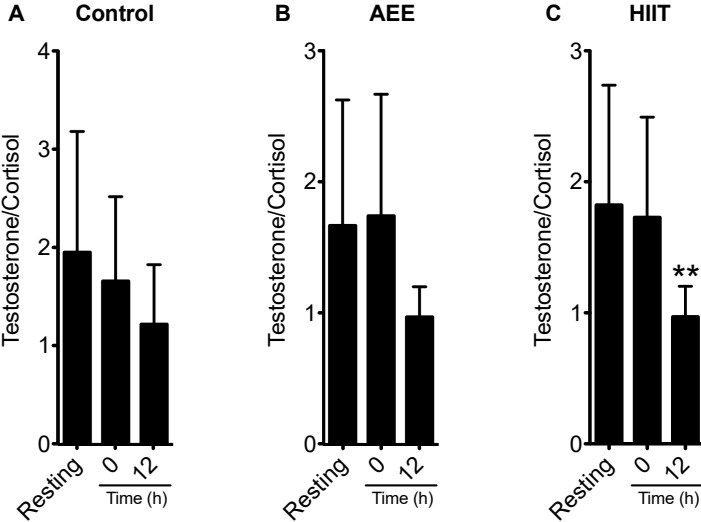

**Figure 4.** Effect of different exercise protocols (AEE and HIIT) on testosterone/cortisol ratio. Testosterone/cortisol ratio (T/C) at rest, immediately after the intervention (0 h) and 12 h after the intervention (12 h), was calculated. (**A**) Control condition; (**B**) AEE condition; (**C**) HIIT condition. ** $p < 0.01$ versus rest. Data shown as mean ± SD.

## 4. Discussion

The aim of this study was to determine the effect of one bout of HIIT and AEE on plasma concentrations of testosterone and cortisol in active male collegiate students. The main results indicated that a HIIT protocol performed at the volume and intensity used does not produce significant differences in hormonal conditions compared to AEE, showing a higher plasma cortisol level 12 h after HIIT compared to AEE.

Testosterone is the major anabolic hormone in men and in contrast, cortisol induces catabolic effects in different tissues, such as skeletal muscle, bone, and heart muscle, among others (1). The T/C ratio is an adequate indicator of the body's anabolic environment [3]. The majority of athletes seek to raise the T/C ratio, thus enhancing protein synthesis and tissue repair following physical activity [16], It's important to mention that protein synthesis can occur without an increase in testosterone. In this study, we observed an increase in cortisol levels 12 h post exercise in response to HIIT exercise, but not after AEE. Surprisingly the control condition also showed an increase in plasma cortisol at 12 h, which could be explained due to plasma cortisol levels oscillating with circadian rhythms reaching a peak in the morning [3]. It is not possible to rule out that the increase in cortisol observed in the HIIT condition was due to the same circadian oscillation phenomenon rather than a direct effect of the physical activity protocol. Studies describe a critical value of the circadian variation of approximately 60% for Cortisol (salivary), somewhat higher than sal-Testosterone (39%) [17,18].

Based on the little evidence available, acute exercise and short-term training do not seem to offer the prospect of altering the circadian profile of T and C to induce some kind of positive adaptation at muscular level and strength [19].

Plasma testosterone levels increased only immediately after exercise, with a similar increase for both AEE and HIIT. Testosterone values in both conditions returned to rest values 12 h post physical activity. In previous studies, it has been reported that testosterone levels vary in response to acute physical exercise, depending on its intensity and duration. For example, testosterone concentration increases after physical stress [20,21] as a result of aerobic activity [22], an increase that is sustained up to 2 h after activity, at which point concentration begins to decrease [22]. These findings would explain the transient increase in testosterone levels observed at time 0 h in both the AEE and HIIT conditions. Contrary to the results observed in this study, Hackney et al. showed that plasma levels of testosterone immediately after exercise have a greater increase in response to HIIT compared with AEE [23], this difference with our findings could be explained by the characteristics of the exercise and the total duration of the HIIT protocol used. In the present work, the total loads applied were lower in volume in both AEE and HIIT compared to the above-mentioned study [22], suggesting that plasma testosterone levels would vary according to the total workload influenced by the total volume (20 vs. 45 min).

It has been described that the testosterone values 12 h after a HIIT protocol are lower than those found at rest (before exercise) and values after AEE [19]. However, these studies have used HIIT exercise protocols that reach fatigue. Exercise protocols ending in fatigue induce an initial increase in plasma testosterone, followed by a marked decrease of this hormone in the hours and days after exercise due to stress [24]. It is noted that the HIIT session within the current study did not produce this marked decrease in testosterone, 12 h after the protocol.

No differences were found in the T/C ratio immediately after AEE or HIIT. Although a reduction in the T/C ratio 12 h post-SIT was observed, it is likely that this reduction is due to the previously discussed circadian oscillation of plasma Cortisol. In addition, although both protocols, AEE and HIIT, increased testosterone levels immediately post-exercise, this increase was not sufficient to modify the T/C ratio, suggesting that these exercise session models alone would not be sufficient to increase the anabolic environment of the organism.

It is considered that sex hormone-binding globulin (SHBG) would not have an effect on the study, previous work by our group concluded that the increase in free testosterone with aerobic exercise was not associated with the change in SHBG binding affinity. In addition, the data suggests that

the exercise-induced increase in testosterone implies increased production that may be mediated by sympathetic stimulation of the testicles [25].

Interestingly, the total workload used in this study is the same as used by Gibala et al. [26] which resulted in an increased VO2 max, which negatively correlates with the risk of cardiovascular death [23]. This last point is relevant because it suggests the possibility of using HIIT exercises to reduce the risk of cardiovascular death and possibly increase cellular anabolism. However, the impact of this exercise modality on populations with special requirements such as diabetics and older adults, among others, as well as the interaction of this type of training with other exercise modalities is still unknown.

## 5. Conclusions

The cortisol concentration 12 h post intervention presented in both control and HIIT condition a significant increase over the pre-intervention state when compared to AEE. The increase in the control condition might be explained by the circadian variation of this hormone.

The testosterone concentrations arose significantly post intervention in both AEE and HIIT condition, but 12 h post intervention there was no difference between conditions, decreasing to baseline (pre-intervention). The T/C ratio decreased (below baseline) significantly only in the HIIT condition 12 h post intervention (below baseline).

**Author Contributions:** C.C.-B. was responsible for conceptualization, funding acquisition, project administration, methodology, data curation, writing—original draft, review and editing. P.R.-L. was responsible for project administration, methodology, writing—original draft, review and editing. T.H.-V. was responsible for conceptualization, methodology, data curation, writing- original draft, review and editing. P.O.-D. was responsible for writing—original draft, review and editing. A.G.-H. was responsible for methodology, data curation, writing—review and editing. A.C.H. was responsible for conceptualization, data curation, writing—review and editing. All authors read and approved the final manuscript.

**Funding:** This research received no external funding.

**Acknowledgments:** To the Vicerrectoria for Research, Development and Innovation of the University of Santiago de Chile. POSTDOC_DICYT, Code: 02164OD, Vicerrectoría de Investigación y Desarrollo, Universidad de Santiago de Chile, USACH, Chile.

**Conflicts of Interest:** The authors declare no conflict of interest.

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
