# Peer review of "Testosterone and Cortisol Responses to HIIT and Continuous Aerobic Exercise in Active Young Men"

_sustainability, doi:10.3390/su11216069_

Round 1

Reviewer 1 Report

Review of Testosterone and Cortisol Responses to HIIT Protocol vs Continuous Aerobic Exercise in Active Young Men

Overall this is an interesting paper and the data needs to be seen by others in the field.

Abstract:

Please include “Thirteen moderately active young adult males…” for clarity to the reader, line 9.

Also indicate that free testosterone was tested since a reader could misread the abstract and believe that it was total testosterone.

Introduction:

Line 28-29, “on the contrary…” this sentence does not read well, please rephrase. “Testosterone is a major anabolic hormone, while cortisol produces generally catabolic effects (citation needed).”

Materials and Methods:

Please include the make and model of the metabolic cart used for VO2 assessment. Also, where standard determination procedures followed? Like respiratory exchange ratio at least 1.1, blood lactate >8mmol/L, etc…

For section 2.3 please state explicitly that blood plasma was measured, this may be unclear to a reader that isn’t aware of EDTA treated tubes.

Please provide more information about the Free testosterone and cortisol assay kits. What brand of plate reader was utilized?

For section 2.4, why was a Dunnet post-hoc utilized rather than a Tukey or Bonferonni?

Results:

I think this is an upload error, but please provide the figures.

Discussion:

Please provide a limitations section. This is a fairly small sample, but the repeated measures design does aid that. However, there didn’t seem to be dietary controls and emotional distress could have been measured through heart rate variability. Also, there is no writing of sex hormone binding globulin, which could really alter the interpretation of this study.

Author Response

Responses Rev. 1

Dear Reviewers, we appreciate your valuable contributions to our article, know that each of your suggestions were considered.

Include "Thirteen moderately active young adult males ..." for clarity for the reader, line 9.

Response: This was incorporated

Also indicate that free testosterone was tested, as a reader could misread the summary and believe it was total testosterone.

Response: This was clarified.

Introduction:

Line 28-29, "on the contrary ..." this sentence does not read well, please rephrase. "Testosterone is an important anabolic hormone, while cortisol generally produces catabolic effects (appointment required)."

Response: This was clarified.

Materials and methods:

Include the brand and model of the metabolic car used for the VO2 evaluation. Also, where were the standard determination procedures followed? As a respiratory exchange ratio at least 1.1, blood lactate> 8 mmol / L, etc.

Response: This was incorporated

For section 2.3, state explicitly that blood plasma was measured, this may not be clear to a reader who does not know the EDTA treated tubes.

Response: This was clarified.

Provide more information on free testosterone and cortisol kits. What brand of plate reader was used?

Response: The analysis was performed in an external laboratory and data on this laboratory were incorporated. (It was not possible to have all the requested information).

For section 2.4, why was a post-hoc Dunnet used instead of a Tukey or Bonferonni?

Responses: This was determined by data characteristics.

Results:

I think this is a loading error, but provide the figures.

Responses: More data and figures were incorporated into results

Discussion:

Provide a section of limitations. This is a fairly small sample, but the design of repeated measures does help. However, there seemed to be no dietary controls and emotional distress could have been measured through heart rate variability. In addition, it has not been written about sex hormone binding globulin, which could really alter the interpretation of this study.

Responses: The request was incorporated.

Reviewer 2 Report

Change to the title from 'Testosterone And Cortisol Response To Hiit Protocol Vs Continuous Aerobic Exercise In Active Young Men' to 'Testosterone And Cortisol Responses To HIIT and Continuous Aerobic Exercise In Active Young Men'.

Abstract:

On first use, high intensity interval training does not require a capital H.

'HIIT protocol exercise' is grammatically incorrect. Please change to 'HIIT'.

Change 'increased similarly plasma levels' to 'similarly increased plasma levels'.

include P values and ES in the abstract. Mean and SD would also be useful for the reader.

Change 'arose' to 'rose'.

Some of the keywords are capitalised and some are not. Please be consistent.

Introduction:

Provide a reference for the first sentence.

line 28: remove 'hormone' from testosterone hormone. 'testosterone' is sufficient.

line 29: The reference 1 does not support these statements. Please provide some mechanistic study that observed T and C have anabolic and catabolic effects respectively.

aerobic exercise does not require capital letters. Likewise high intensity iternal training. 

line 48: provide evidence HIIT is used in trained and untrained individuals.

line 55: reword 'relatively short durations, in efforts and pauses usually less than 1 minute or...'

line 60: this is the first use of 'free testosterone'. Please give some background previously on what free, bioavailable, and total are. The methods section state free testosterone and cortisol (is this total?).

Add a hypothesis.

Methods:

what is the SD for vo2max?

some of the headings have lower-case, and some upper-case. Please be consistent.

Please use the 24 hour clock for reporting time of day (i.e. 19:30 h).

warm up is repeated. Please amend.

'for a fasting the 12 h sample' should read 'for a fasting 12 h sample'.

State what the standardised snack was.

please provide the CV of measurement for hormone analysis. Intra- and inter- assay CV would be useful.

Clarify whether it is total or free cortisol. The phrase 'free testosterone and cortisol' is a bit ambiguous so it would aid transparency to write 'free testosterone and total cortisol' or 'cortisol and free testosterone'.

please add effect size measures into the stats section.

Results:

ml/kg.min should be ml/kg/min or ml.kg.min. The mix of / and . is inconsistent.

There should be reference to table 1 in the text before it is presented.

text should precede the table in the results section.

There is reference to figure 1, but there is no figure in the document.

Include P and ES values in the results.

Discussion:

The third sentence does not make sense. Please reword.

line 139: this reference (1) does not support your statement.

Stating that the T/C ratio leads to increased protein synthesis is misleading. Stu Philipps has shown that protein synthesis can occur without an increase in T. https://www.ncbi.nlm.nih.gov/pubmed/19910330

line 144: Yes, I agree elevations were likely circadian, rather than exercise-induced. Please comment on the criticial difference here (i.e. what change [%] is required before it is biologically meaningful]). https://www.ncbi.nlm.nih.gov/pubmed/25392260

This meta-analysis compares effects of acute exercise modes on salivary T, C, and the T/C ratio (https://www.ncbi.nlm.nih.gov/pubmed/25655373). It could be worthwhile comparing your results to some of the articles within the meta-analysis, even though sal-T and free-T are different, sal-T is purportedly analogous to free-T.

Author Response

Responses Rev. 2

 Dear Reviewers, we appreciate your valuable contributions to our article, know that each of your suggestions were considered.

Summary:

In the first use, high intensity interval training does not require a capital H.

Responses: Modified

The 'HIIT protocol exercise' is grammatically incorrect. Please change to 'HIIT'.

Responses: Modified

Change 'similarly increased plasma levels' to 'similarly increased plasma levels'.

Responses: modified

include values ​​of P and ES in the summary. Mean and SD would also be useful for the reader.

Responses: This was incorporated

Change 'arose' to 'pink'.

Responses: This has changed

Some of the keywords are in uppercase and others are not. Please be consistent.

Responses: This was modified.

Introduction:

Provide a reference for the first sentence.

Responses: The reference was incorporated

line 28: remove 'hormone' from the hormone testosterone. 'testosterone' is enough.

Responses: It was deleted

line 29: Reference 1 does not support these statements. Provide some mechanistic study that observes that T and C have anabolic and catabolic effects respectively.

Responses: This was considered

Aerobic exercise does not require capital letters. Similarly high intensity training.

Responses: This was considered

Line 48: Providing HIIT evidence is used in trained and untrained individuals.

Response: There was no clarity in this suggestion

line 55: reformulate 'relatively short durations, in efforts and breaks generally of less than 1 minute or ...'

Responses: This is clarified in the brief

line 60: this is the first use of 'free testosterone'. Provide some background on what is free, bioavailable and total. The methods section indicates free testosterone and cortisol (is this total?).

Responses: This is clarified under your suggestion

Add a hypothesis.

Responses: Although it is possible and can give clarity, the objective was considered clear.

Methods:

What is the SD for vo2max?

Responses: was incorporated

Some of the headings have lowercase and some uppercase. Please be consistent.

Responses: This was clarified

Use the 24-hour clock to report the time of day (i.e., 7:30 p.m.).

Responses: This was incorporated

The heating is repeated. Please modify

Responses: This was removed

'for a 12-hour fasting sample' I should read 'for a 12-hour fasting sample'.

Responses: This was clarified.

Indicate what the standardized snack was.

Responses: This was incorporated

Provide the measurement CV for hormonal analysis. Intra and inter-trial CV would be useful.

Responses: Data from the laboratory where the analysis was performed was incorporated.

Clarify if it is total or free cortisol. The phrase 'free testosterone and cortisol' is a bit ambiguous, so it would help transparency to write 'free testosterone and total cortisol' or 'free cortisol and testosterone'.

Responses: This was clarified

Add effect size measures in the statistics section.

Responses: This was incorporated

 Results:

ml / kg.min must be ml / kg / min or ml.kg.min. The mixture of / and. is inconsistent

Responses: This was improved

There should be a reference to table 1 in the text before it is submitted.

Responses: This was clarified

the text must precede the table in the results section.

Responses: I indicate where the tables that are attached come

There is a reference to figure 1, but there is no figure in the document.

Responses: This was clarified

Include values ​​of P and ES in the results.

Responses: This was incorporated

Discussion:

The third sentence makes no sense. Please restate.

Responses: This was considered

line 139: this reference (1) is not compatible with your statement.

Responses: This was not well understood.

To affirm that the T / C ratio leads to greater protein synthesis is misleading. Stu Philipps has shown that protein synthesis can occur without an increase in T. https://www.ncbi.nlm.nih.gov/pubmed/19910330

Responses: This was considered and improved in the text

line 144: Yes, I agree that the elevations were probably circadian, rather than induced by the exercise. Comment on the critical difference here (that is, what change [%] is required before it is biologically significant]). https://www.ncbi.nlm.nih.gov/pubmed/25392260

Responses The reference was incorporated and the text was improved

This meta-analysis compares the effects of acute exercise modes on salivary T, C and T / C ratio (https://www.ncbi.nlm.nih.gov/pubmed/25655373). It might be worth comparing your results with some of the articles within the meta-analysis, although salt-T and free-T are different, salt-T is supposedly analogous to free-T.

Responses: the discussion was improved and the reference and discussion articles increased.

Round 2

Reviewer 1 Report

Looks good, thank you for addressing the comments.

Reviewer 2 Report

Kudos on a well-revised manuscript